# Believing that difficulty signals importance improves school outcomes by bolstering academic possible identities, a recursive analysis

Alysia Burbidge[1], Shimin Zhu[2], Sing-Hang Cheung[3], Daphna Oyserman[1]*

**1** Department of Psychology, University of Southern California, Los Angeles, California, United States of America, **2** Department of Applied Social Sciences, The Hong Kong Polytechnic University, Hong Kong, China, **3** Department of Psychology, The University of Hong Kong, Hong Kong, China

* oyserman@usc.edu

**Data Availability Statement:** The surveys (Chinese and the English translation), data, code, and supplemental materials used in this study are available from OSF: https://osf.io/ew3kn/.

## Abstract

Identity-based motivation theory predicts that how sure students are of attaining their academic possible identities (possible identity certainty) interacts recursively with the inferences they make thinking about schoolwork feels hard—this is important for me (difficulty-as-importance) and this is just not for me (difficulty-as-impossibility). Recursivity implies bidirectionality across time points. To date, studies primarily from the U.S. and China only document a shift up or down from one time to a second time. We address this method gap with a three-time-point, two-month study during a secondary school transition (Chinese students N = 818, Mage = 12, 44% female). We obtained prior grades (T0) and placement test scores (T1+). Students filled out questionnaires at critical junctures: a month before the placement test (T1), the day test results were announced (T2), and after learning their school placement (T3). Across three time points, structural equation model results show bidirectional paths linking academic possible identity certainty and difficulty-as-importance beliefs. Controlling for prior grades, students higher in possible identity certainty at T1 scored higher on their placement tests at T1+. Students with higher T1+ placement test scores had higher difficulty-as-importance at T2, controlling for T1 their difficulty-as-importance score. Students with higher T2 difficulty-as-importance had higher T3 possible identity certainty, controlling for their T2 possible identity certainty score. The reverse was also true—students with higher T2 possible identity certainty had higher T3 difficulty-as-importance, controlling for their T2 difficulty-as-importance score. In contrast, controlling for prior scores, from T1 to T2 and T2 to T3 possible identity certainty and difficulty-as-impossibility were unidirectionally related. Students with higher possible identity certainty had lower difficulty-as-impossibility beliefs and not the reverse. Taken together, our results support a recursive process by which difficulty-as-importance beliefs support academic goal pursuit by bolstering students' certainty of attaining their possible identities and certainty reduces difficulty-as-impossibility beliefs.

**Funding:** The author(s) received no specific funding for this work.

## Introduction

Being invested in schoolwork is associated with doing well in school and doing well in school is associated with experiencing schoolwork as self-relevant (for reviews, see [1, 2]; for meta-analyses, see [3–5]). At the same time, self-relevance is not a fixed factor. Instead, according to identity-based motivation (IBM) theory, self-relevance is situation specific. How central something is to one's current or possible selves and how certain one feels about attaining a possible self are context sensitive [6, 7]. When people feel more certain that something is self-relevant, feelings of difficulty experienced while thinking about or working on related tasks and goals can be interpreted as signaling importance and self-relevance [8]. The reverse should also be true. When people interpret difficulty as a signal that a task or goal is important for them, they should feel more certain of attaining a related possible identity. Of course, difficulty can also signal low odds and self-irrelevance. When people feel less certain of an identity, they should be more likely to interpret difficulty as a signal of impossibility, and the reverse. A recent comprehensive review of the voluminous literature on possible future selves and identities [8] documents a variety of conditions under which an accessible future self matters for behavior, including academic outcomes and highlights a dearth of research on the recursive association between people's future identities and the inferences they make from difficulty. To address this gap, we applied IBM theory to examine academic possible identity certainty during an ecologically valid time: the primary-to-secondary school transition. We focus on certainty because research suggests that just having an academic possible identity is not enough to impact academic outcomes [9], but that certainty of attaining academic possible identities is associated with later academic success in both Western and Chinese school samples (for a review, [10]).

### High-stakes school transitions in China and around the world

In China and elsewhere, students can feel pressure to perform academically as secondary school placement is partially determined by test scores and other factors they can influence, as well as by other factors including lotteries and quotas that are not in their control [11–13]. Several countries mandate the use of secondary school placement tests nationwide; even where not mandated, some school systems choose to use secondary school placement tests [14]. For example, in the United States, some regions use high-stakes test scores to allocate scarcely available spots in competitive high schools (e.g., the High School Placement Test in San Francisco and the Specialized High Schools Admissions Test in New York). Hence, though academic stress is not unique to China, the literature from China provides a case study that reveals processes relevant both in this very large population and in other populations as well. Chinese students in urban and rural areas are likely to feel pressure to do well in school, with children as young as 9 to 12 years old reporting high levels of academic stress and fear of punishment for not putting in enough academic effort [15]. Students in China attend six years of primary school and three years of junior secondary school, after which about half continue to senior secondary school and the others enter the workforce directly [16]. Strong placement test scores can help students get from primary school to "key" junior secondary schools that will prepare them for higher education and increase their odds of continuing to senior secondary school and ultimately attending college [17, 18].

### Optimism and accepting fate

While education is valued in a variety of cultures, it is quite central to Chinese culture which emphasizes education as the path to opportunity [19, 20] and fate as predetermining life outcomes [21, 22]. Accepting fate, also termed fatalism, entails a belief that one's life course is in some ways directed by forces beyond one's control and that one's efforts cannot substantially

alter this course [23]. Optimism or optimism for the future entails a belief that no matter how things are in the moment, they will turn out okay or will turn out to have been for the best in the future. Though distinct, the two beliefs might be correlated; believing that the future is fixed does not rule out believing that things will turn out okay [24].

Within Chinese culture, experiencing difficulty is seen as a necessary part of the path to success, an idea described as "eating bitter" [25]. Given this belief that along the way, there is bitterness to be eaten, people can accept their fate [21, 22] while also being generally optimistic about their future possibilities [26] in the face of hardships. Students may apply culturally-available inferences of optimism for the future and accepting fate to make sense of the difficulties they experience during high-stakes school transitions. Empirically, having dispositional optimism for the future is associated with better school outcomes while dispositional fatalism is associated with worse outcomes. Dispositional fatalism is associated with lower academic aspirations among Chinese students [27], fewer academic possible identities among Chinese students [28], less school engagement among Australian students [29], and worse grades among American students [30]. Dispositional optimism for the future is associated with better academic outcomes (meta-analysis of mostly Western college students, [31]; in China, [32]) and having more positive possible identities [33–35]. We did not find studies documenting recursive effects, making it unclear if academic outcomes shape these beliefs, are simply associated with these beliefs, or are shaped by them.

## Identity-based motivation theory: Possible identities and inferences from experiences of metacognitive difficulty

**Overview of identity-based motivation theory.** In this section, we broaden the focus from dispositional optimism and fatalism to consider identity-based motivation theory, which posits a recursive relationship between possible identity certainty and inferences from metacognitive experience of difficulty while thinking. People can feel more or less certain that they have or can attain an identity. Certainty of attaining one's possible identities is expected to vary across situations [6, 7], and this certainty is expected to shape what people infer from their metacognitive experiences of difficulty while thinking about or engaging with self-relevant tasks or goals [8].

Two inferences from metacognitive experiences of difficulty are particularly relevant. First, if a task feels hard to think about or do, people can think 'no pain, no gain, this is valuable for me'–a difficulty-as-importance inference [36]. Second, people can also think 'this task feels hard, maybe it is not for me'–a difficulty-as-impossibility inference [36]. People vary in how much they endorse each inference from difficulty [36, 37]. Moreover, controlling for how much people endorse difficulty-as-importance, higher belief in difficulty-as-impossibility predicts preference for low effort means of goal attainment [38]. In contrast, controlling for how much people endorse difficulty-as-impossibility, higher belief in difficulty-as-importance does not consistently predict preference for easier or more effortful means of goal attainment [38]. The implication may be that dispositional difficulty-as-impossibility is associated with preference for easier means of attaining academic possible identities while dispositional difficulty-as-importance is associated with more certainty of attaining academic possible identities.

Whether difficulty-as-importance beliefs and certainty of attaining possible identities recursively shape each other across time is not yet clear. Yet, identity-based motivation theory posits a recursive relationship over time such that people who feel certain they can attain a possible identity should endorse difficulty-as-importance when thinking about related tasks or goals feels hard. They should feel more certain they can attain their possible identities and this certainty should carry over to stronger belief in difficulty-as-importance. Then, when schoolwork

feels hard to think about or do, higher difficulty-as-importance beliefs should carry forward to more certainty that they can attain their academic possible identities. Continuing the recursive chain, this certainty should carry over to further bolster their difficulty-as-importance beliefs.

**Applying identity-based motivation theory to make sense of the literature connecting future self to current action.**   A 2023 comprehensive review of all the experimental and correlational studies published between 1985 and 2022 on the link between future selves and behavior reveals three competing assumptions about how the future self matters for behavior, each supported by its own siloed subset of the literature [8]. The first siloed subset of the literature focuses on self-continuity and shows effects of including the future self in the current self. The second siloed subset of the literature focuses on self-contrast and shows effects of feeling efficacious about overcoming the gap between the current and the future self. The third siloed subset of the literature focuses on other aspects of future selves including their valence and link to strategies and shows effects of holding a positively or a negatively valenced future self and of having strategies linked to the possible self. Each of the three siloed literatures provides supporting evidence for its claim but cannot explain why supporting evidence also exists for the alternative claims. If what matters is self-continuity (i.e., including one's future self in the current self), why would imagining a gap between the current and future self and feeling efficacious about overcoming that gap sometimes trigger future-focused behavior? Similarly, if either self-continuity or self-contrast is what matters, why would features of the imagined future self (i.e., a possible selves model) sometimes trigger future-focused behavior? After looking for and not finding explanations rooted in sample, research method, or dependent measure differences that could clarify these competing results, the review authors suggest considering unmeasured mediators or moderators. In particular, they focus on the inferences people draw from their metacognitive experience of difficulty [8]. When thinking about one's future self feels hard, people may feel less certain they can attain it (difficulty-as-impossibility). When thinking about one's future self feels hard but one also feels certain they can attain that future self, then they may infer that self-relevant behaviors are important to engage in [8].

**Summary of difficulty-as-importance and difficulty-as-impossibility research.**   Experimental studies testing the unidirectional effects of endorsing or having accessible a difficulty-as-importance or difficulty-as-impossibility inference find results in line with this proposal. When researchers vary whether students have an accessible difficulty-as-importance or difficulty-as-impossibility inference on the mind, they find the expected shifts in students' certainty of attaining their academic possible identities [39, 40]. Students with difficulty-as-impossibility on the mind report that a difficult standardized test was harder and performed worse on it than students with difficulty-as-importance on the mind [41]. Research examining the association between students' reported boredom during demanding test situations and poorer test performance suggests that American students (and teachers) may lack the vocabulary to describe their inferences from metacognitive experiences of difficulty [42]. If given the terminology, rather than saying that demanding tasks are boring, they may instead identify the response as difficulty-as-impossibility–the sense that it is not for them, a waste of time, and hence something not worth engaging in.

Beliefs in difficulty-as-importance and difficulty-as-impossibility may be differentially accessible across countries, as evidenced in comparisons between people from the U.S., India, and China [37]. Cross-cultural comparisons suggest that people from Western countries are more likely to endorse difficulty-as-impossibility, though no more likely to endorse difficulty-as-importance, than people from China, India, Iran, and Turkey [43]. A number of studies examined the discriminant validity and measurement invariance of the difficulty-as-importance and difficulty-as-impossibility scales, reporting evidence of measurement invariance in

samples from the U.S., other Western countries, and less Westernized countries like China, Iran, India, and Turkey [37, 43].

Several studies also provide evidence that difficulty-as-importance and difficulty-as-impossibility beliefs are distinct, not flip sides of the same construct [36, 37]. For example, people who score higher in difficulty-as-importance are more likely to believe that they can make time for important tasks and goals, controlling for their difficulty-as-impossibility scores [40]. The same is true when comparing people with difficulty-as-importance to those with difficulty-as-impossibility on the mind [44]. Difficulty-as-importance beliefs are distinct from other self-beliefs; they are not redundant nor reliably associated with optimism for the future [43], self-esteem [45], or endorsing a fixed mindset [36, 37]. Controlling for people's endorsements of difficulty-as-importance, higher difficulty-as-impossibility endorsers score lower on optimism for the future [43], report lower self-esteem [45], and are more likely to endorse a fixed mindset [36, 37]. We did not find prior work examining the relationship between fatalism and endorsing either difficulty-as-importance or difficulty-as-impossibility.

## Gaps to be addressed

Evidence to date documents pieces of the posited recursive process but has not tested the full recursive process between possible identity certainty, academic outcomes, and the difficulty-as-importance and difficulty-as-impossibility inferences students draw from their metacognitive experiences of difficulty. Specifically, studies document paths from difficulty-as-importance and difficulty-as-impossibility to identity certainty [39, 40] and from identity certainty to school outcomes [10], between possible identities and accepting fate [28], and between optimism and difficulty-as-impossibility [43]. In the current study, we address this methodological gap, making progress beyond prior piecemeal testing by testing the posited recursive relationships. Specifically, we draw from identity-based motivation theory and studies suggesting that dispositional optimism and fatalism matter for academic outcomes and apply this to a high-stakes school transition.

## The current study

Given the recursive predictions posited by identity-based motivation theory, we explore bidirectional relationships in the current study. We measure school achievement, students' certainty of attaining their academic possible identities, their difficulty-as-importance and difficulty-as-impossibility inferences from their metacognitive experiences of difficulty related to school tasks, their optimism for the future, and their fatalism about how the future will unfold. Because recursivity requires a temporal ordering, our data with multiple time points provides a useful first test. To study recursivity, the analysis includes controlling for the temporally prior assessment of achievement and of the self-belief that is the predicted outcome in each step of the analysis–academic possible identity certainty, difficulty-as-importance, and difficulty-as-impossibility, optimism for the future, accepting fate. This allows us to examine the temporally-lagged effects of changes in achievement and changes in self-beliefs on trajectories of achievement and self-beliefs over the school transition.

We present our synthesized process model in Fig 1. RQ1 addresses the top panel of Fig 1, the recursive relationship between students' certainty of attaining their academic possible identities and their academic outcomes. RQ2 addresses the bottom panel of Fig 1. It explores recursive relationships between students' certainty of attaining their academic possible identities and their self-beliefs about difficulty-as-importance, difficulty-as-impossibility, optimism for the future, and fatalism about how the future will unfold. These analyses consider both the predicted recursive relationships between certainty and difficulty inferences posited by

Possible Identity Certainty Model

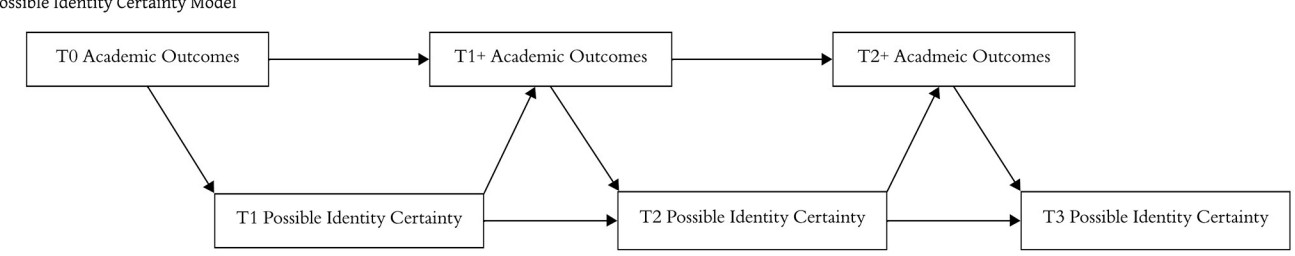

Mediated Process Model

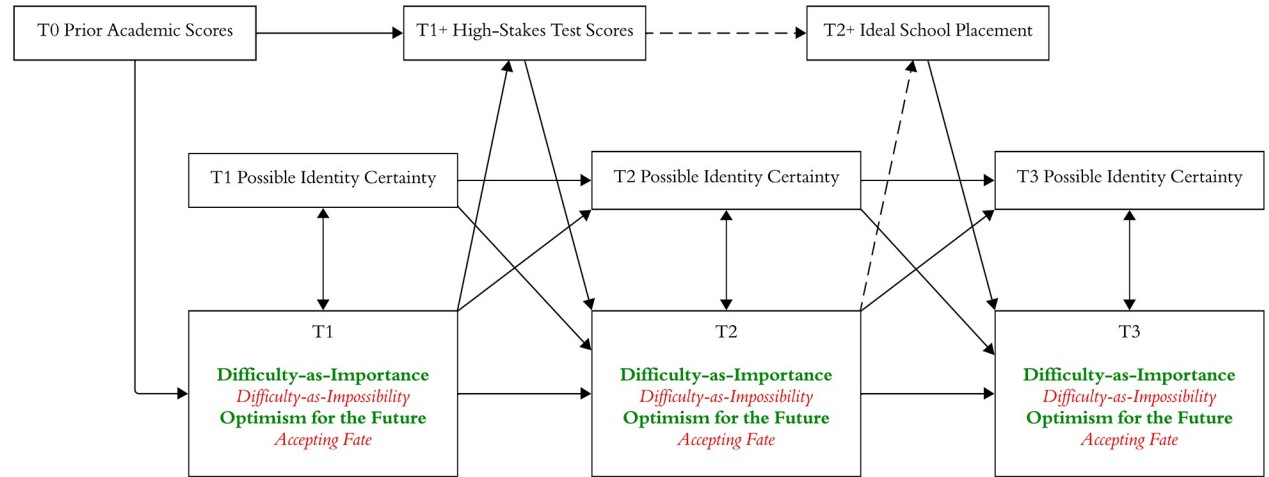

**Fig 1. Theoretical process models.** Green bolded text = association predicted to be positive. Red italicized text = association predicted to be negative. Dashed line = association predicted to be weak.

identity-based motivation theory and the additional potential recursive relationships between certainty and optimism for the future and fatalistic acceptance of it.

Fig 2 presents the flow of our data collection from T0 to T3. We collected T1 to T3 data at three time points during a two-month segment as Chinese primary school students transitioned to secondary school. This allowed us to leverage multiple time points during a naturally occurring temporal process. First, we obtained prior semester academic grades, labeled as T0. We label as T1 the first time students completed a survey, a month prior to their high-stakes secondary placement test. We label the test itself as T1+. A week after the test, students learned their test scores and completed the second survey; we label this as T2. Four weeks after the test,

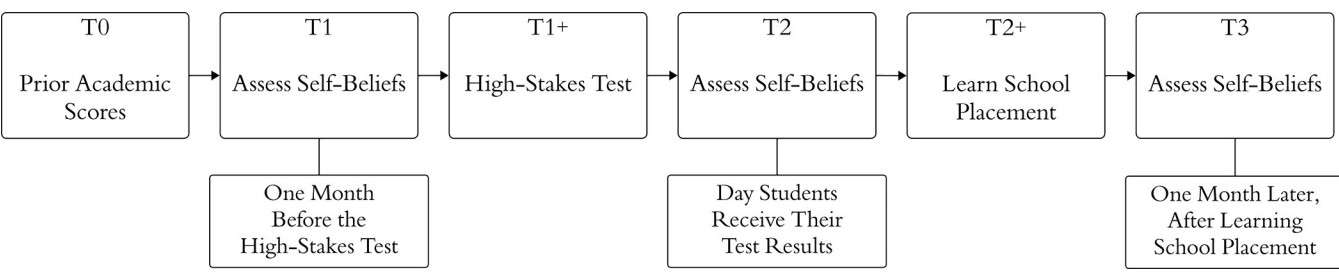

**Fig 2. Data collection timeline.**

they completed the third survey; we label this as T3. Students learned their school placement between T2 and T3; we label this T2+.

## Sample and method

### Ethics statement

We obtained IRB approval from the University of Hong Kong's Human Research Ethics Committee for Non-Clinical Faculties. Participation was voluntary. Once schools agreed to participate, we mailed parents a consent form and a study information sheet. Students could sign assent forms if their parents returned signed consent forms.

### Participants

Participants were twelve-year-old students (N = 818, 44% female) attending three elementary schools in an urban area of Guangdong Province, China. We obtained age and gender information at T1. As shown in Table 1, across T1 to T3, demographic profiles remained stable and the three schools provided about equally to the sample. Compared to our participant roster, the response rate was 94% at T1, 92% at T2, and 60% at T3. We do not have further information on features of the schools and do not use school as a variable in further analyses. As detailed in the Descriptive Measures section, we obtained a snapshot at T1 from participants of their daily activities and effort in school. This descriptive snapshot is presented in our Supplemental Materials, S1 to S3 Tables in S1 File.

### Procedure

Before starting data collection, we obtained university IRB approval as part of a seed grant. Participation was voluntary. Students participated if their parents signed consent forms and students signed consent forms. Students received a token (pencil case) for each time survey they completed. Students lived in a middle-sized Chinese city with 15 primary schools. To obtain an adequate sample size, we recruited three schools by sending recruitment letters one by one to randomly selected schools, sending five letters in total. The letters invited schools to let their 6th-graders participate in a study examining the transition from primary school to secondary school. To the best of our knowledge, accepting and declining schools did not differ in neighborhood characteristics. Declining schools could not fit our study into their school calendar and management.

We coordinated data collection so that two research assistants visited each school on the same day and asked students to fill out surveys. Teachers were not in the classroom during data collection. The alternate activity was to sit and read either in the classroom or in a room

**Table 1. T1 Age, % female, and proportion of sample in each school at each time point.**

|  | Time 1 | Time 2 | Time 3 |
|---|---|---|---|
| Mean Age (SD) at T1 | 12.28 (0.61) | 12.26 (0.61) | 12.27 (0.60) |
| % Female at T1 | 44.10 | 44.47 | 44.44 |
| % of Total Sample by School |  |  |  |
| School A | 42.75 | 43.43 | 38.54 |
| School B | 23.45 | 21.51 | 22.92 |
| School C | 33.81 | 35.06 | 38.54 |

SD = standard deviation. We asked for age and gender at Time 1 only. Slight variability in the mean and SD of age and % female across time points is a function of missing data across time points.

outside the classroom. Students filled in their T1 surveys a month prior to the high-stakes test (T1+) on a normal school day. A week after the test, students came to school in the morning to receive their test results and could return that afternoon to collect their transcripts from their classrooms, at which time research assistants asked them to complete the T2 survey. Four weeks later, students learned their school placement and could return to their classrooms during their summer break to collect their graduation certificates, at which time research assistants asked them to complete the T3 survey. We obtained age and gender data only at T1.

## Data transparency

Our data, code, and supplemental materials (including the Chinese and English translations of our scales) are in OSF: https://osf.io/ew3kn/?view_only=52386fbc3f3a4b5db666a0fcaa07b348. We also share our supplemental materials in the Supporting Information S1 File.

## Inclusivity in global research

Additional information regarding the ethical, cultural, and scientific considerations specific to inclusivity in global research is included in the Supporting Information S2 File.

# Measures

Measures were written to be understandable to primary school students. We used a translation-back translation method [46] for scales in English. Primary measures were collected at T1, T2, and T3. Descriptive measures were collected at T1 or T2.

## Descriptive measures

To provide a snapshot of the sample, at T1, we asked students to estimate the time they typically spent on daily activities, choose their favorite subject from among mathematics, Chinese, English, or "Other", rate how much effort they were exerting in $6^{th}$ grade (1 = not working hard, 10 = working very hard), and respond to three items about their secondary school placement. Students rated how likely (0 = not likely at all, 9 = very likely) it was that they would get placed in the secondary school they preferred ("ideal school"), how hopeful (0 = not hopeful at all, 9 = very hopeful) and how worried (0 = not at all worried, 9 = very worried) they were about placement. Given that some schools were more academically competitive than others, we thought that this might be a way of assessing their subjective belief about their academic chances. At T2, we asked students to what extent they felt their placement test score was a measure of their true ability, effort, luck, and fate (0 = not at all true, 9 = completely true). We share descriptive results in Supplemental Materials, S1 to S3 Tables in S1 File.

## Primary measures

**Academic outcomes.**   Schools shared prior semester scores (T0 grades, M = 81.5%; raw scale 0–300, raw M = 244.50 SD = 41.25) and high stakes placement test scores (T1+ test scores, M = 84.33%; raw scale 0–300, raw M = 253.00, SD = 39.87) with us. Both scores were a combination of Chinese, English, and Math subject tests (upper bound of 300 points). At T1, students named the secondary school they hoped to attend (ideal school). At T3, they indicated whether their school placement was their ideal school (1 = yes, 2 = no, 60.15% yes).

**Certainty of attaining academic possible identities.**   Students rated how certain (0 = not possible at all to 9 = very possible) they were of attaining their academic possible identities in the coming year with Kemmelmeier and Oyserman's 3-item scale [47] ("I will perform well in

**Table 2. Scale means (M), standard deviations (SD), and α reliability at Time 1, Time 2, and Time 3.**

| Scale | Time 1 | | | Time 2 | | | Time 3 | | |
|---|---|---|---|---|---|---|---|---|---|
| | M | SD | α | M | SD | α | M | SD | α |
| Difficulty-as-Importance | 4.27 | 0.65 | .86 | 4.23 | 0.69 | .88 | 4.11 | 0.74 | .91 |
| Difficulty-as-Impossibility | 1.75 | 0.72 | .85 | 1.78 | 0.76 | .88 | 1.86 | 0.73 | .90 |
| Accepting Fate | 1.82 | 0.73 | .86 | 1.93 | 0.85 | .92 | 1.95 | 0.83 | .93 |
| Optimism for the Future | 3.56 | 0.84 | .79 | 3.50 | 0.92 | .82 | 3.59 | 0.90 | .87 |
| Academic Possible Identity Certainty | 6.72 | 1.59 | .86 | 6.80 | 1.72 | .89 | 6.98 | 1.61 | .91 |

All scales are 1 = strongly disagree, 5 = strongly agree, except for academic possible identity certainty which is 0 = not possible at all, 9 = very possible. Statistics are based on finalized measures (for details, see Results: Construct Validity).

school", "I will have good grades", "I will understand the materials in class". Other researchers conducting studies with Chinese samples use variations of Kemmelmeier and Oyserman's scale (e.g., [48], or as a single item indicator [49], a score generated from qualitative analyses of diary entries [50], or an open-ended measure of possible identities [28]). As shown in Table 2, students scored above the midpoint on possible identity certainty and αs ranged from .86 to .91. We also had students rate their certainty of using effective strategies to attain these possible identities with Kemmelmeier and Oyserman's 4-item scale [47] but, as detailed in the preliminary analysis section, we did not use this scale in further analyses.

**Accepting fate.** Students rated how much they agreed or disagreed (1 = strongly disagree, 5 = strongly agree) with the 8-item Chinese-language Adolescent Fatalism Questionnaire [28] (e.g., "I have not realized my ideals, mainly because it is not my fate to do so.") and two added exploratory items. As detailed in the Results: Construct Validity section, we did not use these two exploratory items in further analyses. As shown in Table 2, students scored above the midpoint on accepting fate and αs ranged from .86 to .93.

**Optimism for the future.** Students rated how much they agreed or disagreed (1 = strongly disagree, 5 = strongly agree) with a 7-item Life Orientation Test-Revised scale [51] (e.g., reverse-scored "I feel like good things rarely happen to me."). When used in Chinese translation, researchers such as [52] find that negatively worded items like the item in the above example do not scale with positively worded ones (e.g., "I am always optimistic about my future."). We adapted two items for use with children and, as described in our Results: Construct Validity section, also found that only negatively worded items load together. As shown in Table 2, using the four negatively worded items as our scale, students scored above the midpoint on optimism for the future and αs ranged from .79 to .87.

**Difficulty-as-importance and difficulty-as-impossibility.** Students rated how much they agreed or disagreed (1 = strongly disagree, 5 = strongly agree) with conceptual translations of the 6-item Oyserman and colleagues' difficulty-as-importance scale [53] (e.g., "To achieve good student results, there is no doubt that real hard work is required. Those things that require me to really work hard mean these things are important to me.") and difficulty-as-impossibility scale [53] (e.g., "Sometimes, it feels too difficult to learn—it is even impossible to learn well. This may be a good thing because it makes me understand that I should do other things."). The conceptual translations concretized school tasks as learning, studying, reading, and test taking. As detailed in the Results: Construct Validity section, we dropped one item from the difficulty-as-impossibility scale. As shown in Table 2, students scored above the midpoint on difficulty-as-importance and below the midpoint on difficulty-as-impossibility; αs for the former ranged from .86 to .91 and for the latter ranged from .85 to .90.

## Analysis plan

We conducted our preliminary and main analyses using the statistical software R [54]. For our preliminary analysis, we tested the construct validity of our primary measures, verified longitudinal measurement invariance for each measure, and described our sample. For our main analysis, we used a first-order structural equation modeling (SEM) approach to model temporal processes and control for temporally prior scores for each of the repeated measures delineated in Fig 1.

First, to address RQ1, we modeled the temporal and bidirectional relationships between academic outcomes and certainty of attaining academic possible identities. Then, to address RQ2, we added the theory-based recursive paths for the other four self-beliefs: difficulty-as-importance, difficulty-as-impossibility, optimism for the future, and accepting fate. We controlled for prior grades when predicting high-stakes test scores and for test scores when predicting T1 self-beliefs, for T1 self-beliefs when predicting T2 self-beliefs, and for T2 self-beliefs when predicting T3 self-beliefs. We did not use T0 grades or T1 + test scores as controls when predicting T2+ ideal school placement (whether or not students were placed in the school they hoped to get into) in our SEM models given that the T0 grades and T1+ test scores had very low correlations with this T2+ variable, as described below.

Throughout, we chose to use first-order SEM models, as second-order models would require estimating the value of the latent variables. This would decrease the degrees of freedom and necessitate a substantially larger sample than ours to detect significant effects. We were well-powered for first-order models. A post hoc power analysis using G*Power [55] determined our analytic sample size (N = 818) had sufficient power of .82 to detect small effect sizes ($f^2$ = 0.02) with an alpha of .05 in our most complex paths (e.g., a path with nine predictors) and power of .96 in our simplest paths (e.g., a path with two predictors). We used maximum likelihood estimation with robust (Huber-White) $SE$ to estimate each SEM model.

For a number of reasons, other modeling approaches such as cross-lagged panel models (CLPMs) were not appropriately suited for the current study. First, our ecologically valid data collection timeline (Fig 2) meant that our data structure did not meet the assumption of synchronicity for cross-lagged panel model (CLPM) approaches [56]. Second, our focus on the high-stakes testing period meant our third school outcome data point is the match between school placement and student preference. A third academic score data point would be needed for a random intercept CLPM [57]. Third, a CLPM would not allow us to simultaneously evaluate how the school placement match, a one-time outcome, may affect subsequent self-beliefs.

We imputed missing data (see Supplemental Materials S4 Table in S1 File for the proportion of missing data for each analytic variable) using the R mice package (Multivariate Imputation by Chained Equations [58]). Multivariate imputation by chained equations can appropriately impute missing values when both dependent variables and independent variables have missing data points and data is missing at random [59, 60]. Multiple imputation restores the natural variability in the missing data while accounting for the uncertainty caused by estimating missing data with consideration to how variables in the analytic dataset correlate with the missing data [61]. As per [62] and [63], we imputed missing values at the item-level and created unique multiple imputation models for each missing variable. Following [63], we used a mixed item-scale approach to determine predictors of each multiple imputation model. We imputed only variables in our SEM analysis, not descriptive variables.

## Results

### Preliminary analysis

**Construct validity.**   Confirmatory factor analysis results showed that academic possible identity certainty and certainty of using effective strategies to attain these possible identities were correlated but separate constructs. We attempted to include the latter in our SEM model but this resulted in a 33% increase in AIC and BIC statistics, indicating the model's increased complexity was not counterbalanced by increased model fit. Hence, we did not include certainty of using academic strategies in further analyses. We also used confirmatory factor analysis to confirm our theory-based decision to model each self-belief measure as a separate construct (for detailed results, see Supplemental Materials S5 and S6 Tables in S1 File). The confirmatory factor analysis revealed five poor-loading items (<0.6) that we dropped to ensure acceptable fit for our analytic model. The five dropped items were one of the six items in the difficulty-as-impossibility scale, the two exploratory items we had added to the 8-item accepting fate scale, and the three positively valenced optimism items (one adapted, two originals), yielding a 4-item optimism for the future scale. Measures were reliable with Cronbach's alpha coefficients ranging from .79 to .93 (Table 2).

We examined the associations among the three academic outcome measures, finding that T0 grades and T1+ high-stakes test results were correlated highly ($r = .93$, $p < .001$). Each had only a small correlation with the third measure, students' T2+ report that they were placed in the school they hoped to get in ($r = .08$, $p = .031$ with T0 grades and $r = .12$, $p = .001$ with T1 + test results). We infer that the two objective assessments of performance were only weakly driving where students wanted to continue their education. We examined the associations among the self-belief measures. Certainty of attaining academic possible identities correlated moderately with difficulty-as-importance, difficulty-as-impossibility, optimism for the future, and accepting fate ($|.19|<rs<|.37|$, $ps < .001$). Correlations among difficulty-as-importance, difficulty-as-impossibility, optimism, and accepting fate were moderate at each time point, $|.26|<rs<|.56|$, $ps < .001$, and in the expected directions. As detailed in Table 3, we found positive point in time associations between accepting fate and difficulty-as-impossibility and between difficulty-as-importance and optimism for the future, and negative associations between difficulty-as-impossibility and difficulty-as-importance and accepting fate and optimism for the future.

**Repeated measures: Stability and invariance.**   As detailed in Table 4, between T1 and T2 and between T2 and T3, each repeated measure correlated strongly with itself ($.62<rs < .74$, $ps < .001$).

**Table 3. Bivariate correlations of the self and motivation scales at each time point.**

| Scale | Time 1 | | | | Time 2 | | | | Time 3 | | | |
|---|---|---|---|---|---|---|---|---|---|---|---|---|
| | 2 | 3 | 4 | 5 | 2 | 3 | 4 | 5 | 2 | 3 | 4 | 5 |
| 1 | -.45 | -.28 | .26 | .37 | -.50 | -.30 | .26 | .30 | -.49 | -.30 | .29 | .30 |
| 2 | – | .47 | -.43 | -.34 | – | .56 | -.40 | -.30 | – | .55 | -.45 | -.26 |
| 3 | | – | -.48 | -.27 | | – | -.49 | -.19 | | – | -.42 | -.21 |
| 4 | | | – | .31 | | | – | .27 | | | – | .19 |
| 5 | | | | – | | | | – | | | | – |

1 = Difficulty-as-Importance, 2 = Difficulty-as-Impossibility, 3 = Accepting Fate, 4 = Optimism for the Future, 5 = Academic Possible Identity Certainty. All correlations $p < .001$.

**Table 4. Intra-measure correlations between T1 and T2 and between T2 and T3.**

| Scale | Correlations Between | |
|---|---|---|
| | T1 and T2 | T2 and T3 |
| Difficulty-as-Importance | .59 | .58 |
| Difficulty-as-Impossibility | .64 | .65 |
| Accepting Fate | .63 | .71 |
| Optimism for the Future | .65 | .64 |
| Certainty of Attaining Academic Possible Identities | .66 | .73 |

All correlations $p < .001$.

We explored consistency in each measure's internal structure over time using longitudinal measurement invariance analyses (Supplemental Materials S7 Table in S1 File). Each configural model had at least acceptable fit, with CFI>.90, SRMR < .08, RMSEA < .08 [64, 65]. Following [66], we determined invariance via changes in CFI supplemented by changes in SRMR and RMSEA. We found metric and scalar invariance for each self-belief. These results suggest our measures are internally stable over time and so repeated measurements can be meaningfully compared.

## Main analysis

**RQ1: Academic outcomes and certainty of attaining one's academic possible identities (possible identity certainty).** We tested the relationship between students' academic outcomes and how certain they were about attaining their academic possible identities (possible identity certainty). We share complete path statistics in Table 5. The model had adequate fit: CFI = .99, RMSEA = .07 (90% CI [.04, .10]), SRMR = .02. At T1, students scored higher in possible identity certainty if they had better T0 grades ($B = 1.43$, $z = 11.28$, $p < .001$). Moreover, controlling for their T0 grades, students who at T1 felt more certain about attaining their academic possible identities scored higher at T1+ on their high-stakes test ($B = 0.01$, $z = 2.49$, $p = .013$). Controlling for their T1 possible identity certainty, students who scored higher on their T1+ high-stakes test scores felt more certain about attaining the academic possible identities at T2 ($B = 0.93$, $z = 5.89$, $p < .001$). Controlling for their T1 and T2 possible identity certainty scores, students' T3 possible identity certainty scores were not associated with whether their T2+ secondary school placement matched the one they hoped to get into.

**Table 5. Path analysis RQ1: Academic outcomes and academic possible identity certainty.**

| Outcome Variable | Predictor Variable | B | SE B | β | z | p | R² |
|---|---|---|---|---|---|---|---|
| T1 Possible Identity Certainty | T0 Prior Academic Scores | 1.43 | 0.13 | 0.37 | 11.28 | < .001 | .14 |
| T1+ High-Stakes Test Scores | T0 Prior Academic Scores | 0.89 | 0.02 | 0.92 | 47.79 | < .001 | .87 |
| | T1 Possible Identity Certainty | 0.01 | 0.00 | 0.04 | 2.49 | .013 | |
| T2 Possible Identity Certainty | T1 Possible Identity Certainty | 0.63 | 0.04 | 0.58 | 18.11 | < .001 | .47 |
| | T1+ High-Stakes Test Scores | 0.93 | 0.16 | 0.22 | 5.89 | < .001 | |
| T3 Possible Identity Certainty | T1 Possible Identity Certainty | 0.29 | 0.03 | 0.28 | 8.48 | < .001 | .57 |
| | T2 Possible Identity Certainty | 0.50 | 0.03 | 0.54 | 15.20 | < .001 | |
| | T2+ School Placement | -0.07 | 0.08 | -0.02 | -0.95 | .344 | |

$B$ = unstandardized coefficient. $SE\ B$ = standard error of unstandardized coefficient. β = standardized coefficient.

**RQ2: Adding difficulty-as-importance, difficulty-as-impossibility, optimism, and accepting fate.** The model had adequate fit: CFI = .96, RMSEA = .08 (90% CI [.07, .08]), SRMR = .08. We detail the recursive process next, piece by piece. We share complete path statistics in Table 6 and the visualization of all results in Fig 3.

T1 difficulty-as-importance, difficulty-as-impossibility, optimism for the future, and accepting fate were the outcome variables in the first path which explored the effect of T0 grades on these four self-beliefs at T1. T0 grades were associated with each of these T1 constructs. Students who had better T0 grades had higher difficulty-as-importance ($B = 0.44$, $z = 7.15$, $p < .001$) and optimism for the future scores ($B = 0.46$, $z = 6.49$, $p < .001$) and lower difficulty-as-impossibility ($B = -0.62$, $z = -9.50$, $p < .001$) and accepting fate scores ($B = -0.55$, z = -8.22, $p < .001$).

T1 difficulty-as-importance, difficulty-as-impossibility, optimism for the future, and accepting fate were the predictor variables in the second path. The second path explored their T1 effects on T1+ high-stakes test scores. Accounting for the strong effects of T0 grades and T1 academic possible identity certainty, T1 difficulty-as-importance, difficulty-as-impossibility, optimism, and accepting fate were not significantly associated with T1+ high-stakes test scores. T1 difficulty-as-importance, difficulty-as-impossibility, optimism for the future, and accepting fate were also the predictor variables in the third path. The third path explored their effects on T2 possible identity certainty, accounting for the effects of T1 possible identity certainty and T1+ high-stakes test scores. T2 possible identity certainty was higher among students whose T1 difficulty-as-importance ($B = 0.18$, $z = 2.02$, $p = .043$) and optimism for the future scores ($B = 0.22$, $z = 3.36$, $p = .001$) were higher; their T1 difficulty-as-impossibility and accepting fate scores did not matter (no significant associations).

T2 difficulty-as-importance, difficulty-as-impossibility, optimism for the future, and accepting fate were the outcome variables in the fourth path. The fourth path explored the effects of T1+ high-stakes test scores and T1 possible identity certainty on T2 difficulty-as-importance, difficulty-as-impossibility, optimism for the future, and accepting fate. T2 occurred about one month after T1, on the day students learned their high-stakes test results. Accounting for their T1 difficulty-as-importance scores, students' T2 difficulty-as-importance scores were higher if at T1 they felt more certain about attaining their academic possible identities ($B = 0.04$, $z = 2.45$, $p = .014$) and at T1+ did better on their high-stakes test ($B = 0.12$, $z = 2.10$, $p = .036$). The reverse was true for difficulty-as-impossibility and accepting fate scores. Accounting for their T1 difficulty-as-impossibility scores, students' T2 difficulty-as-impossibility scores were lower if at T1 they felt more certain about attaining their academic possible identities ($B = -0.03$, $z = -2.21$, $p = .027$) and at T1+ they did better on their high-stakes test ($B = -0.13$, $z = -2.28$, $p = .023$). Regarding accepting fate, accounting for their T1 accepting fate scores, students' T2 accepting fate scores were lower if at T1+ they did better on their high-stakes test ($B = -0.21$, $z = -3.08$, $p = .002$); their T1 possible identity certainty score did not matter (no significant association). Lastly, regarding T2 optimism for the future, accounting for students' T1 optimism scores, neither their T1 possible identity certainty score nor their T1 + high-stakes test scores mattered (no significant associations). In sum, both T1 possible identity certainty scores and T1+ high-stakes test scores mattered for T2 difficulty-as-importance and difficulty-as-impossibility scores, only T1+ high-stakes test scores and not T1 possible identity certainty scores mattered for T2 accepting fate, and neither T1+ high-stakes test scores nor T1 possible identity certainty mattered for T2 optimism for the future.

T2 difficulty-as-importance, difficulty-as-impossibility, optimism for the future, and accepting fate were the predictor variables in the fifth path. The fifth path explored the effects of these four self-beliefs on T3 possible identity certainty. T3 was about one month after T2, when students knew the secondary school they would attend in the fall. Accounting for their

**Table 6. Path analysis RQ2: Academic Outcomes, academic possible identity certainty, difficulty-as-importance, difficulty-as-impossibility, optimism for the future, and accepting fate.**

| Outcome Variable | Predictor Variable | B | SE B | β | z | p | $R^2$ |
|---|---|---|---|---|---|---|---|
| T1 Possible Identity Certainty | T0 Prior Academic Scores | 1.43 | 0.13 | 0.37 | 11.28 | < .001 | .14 |
| T1 Difficulty-as-Importance | T0 Prior Academic Scores | 0.44 | 0.06 | 0.28 | 7.15 | < .001 | .08 |
| T1 Difficulty-as-Impossibility | T0 Prior Academic Scores | -0.62 | 0.07 | -0.36 | -9.50 | < .001 | .13 |
| T1 Accepting Fate | T0 Prior Academic Scores | -0.55 | 0.07 | -0.31 | -8.22 | < .001 | .10 |
| T1 Optimism for the Future | T0 Prior Academic Scores | 0.46 | 0.07 | 0.23 | 6.49 | < .001 | .05 |
| T1+ High-Stakes Test Scores | T0 Prior Academic Scores | 0.88 | 0.02 | 0.91 | 43.77 | < .001 | .87 |
|  | T1 Possible Identity Certainty | 0.01 | 0.00 | 0.03 | 1.81 | .071 |  |
|  | T1 Difficulty-as-Importance | 0.00 | 0.01 | 0.01 | 0.43 | .671 |  |
|  | T1 Difficulty-as-Impossibility | -0.01 | 0.01 | -0.03 | -1.23 | .218 |  |
|  | T1 Accepting Fate | -0.01 | 0.01 | -0.02 | -0.88 | .378 |  |
|  | T1 Optimism for the Future | -0.01 | 0.01 | -0.01 | -0.53 | .598 |  |
| T2 Possible Identity Certainty | T1 Possible Identity Certainty | 0.58 | 0.04 | 0.54 | 15.49 | < .001 | .49 |
|  | T1+ High-Stakes Test Scores | 0.88 | 0.17 | 0.20 | 5.23 | < .001 |  |
|  | T1 Difficulty-as-Importance | 0.18 | 0.09 | 0.07 | 2.02 | .043 |  |
|  | T1 Difficulty-as-Impossibility | 0.04 | 0.09 | 0.02 | 0.51 | .608 |  |
|  | T1 Accepting Fate | 0.07 | 0.08 | 0.03 | 0.84 | .403 |  |
|  | T1 Optimism for the Future | 0.22 | 0.07 | 0.11 | 3.36 | .001 |  |
| T2 Difficulty-as-Importance | T1 Difficulty-as-Importance | 0.53 | 0.05 | 0.52 | 11.86 | < .001 | .33 |
|  | T1 Possible Identity Certainty | 0.04 | 0.01 | 0.08 | 2.45 | .014 |  |
|  | T1+ High-Stakes Test Scores | 0.12 | 0.06 | 0.07 | 2.10 | .036 |  |
| T2 Difficulty-as-Impossibility | T1 Difficulty-as-Impossibility | 0.54 | 0.03 | 0.53 | 16.04 | < .001 | .35 |
|  | T1 Possible Identity Certainty | -0.03 | 0.02 | -0.07 | -2.21 | .027 |  |
|  | T1+ High Stakes Test Scores | -0.13 | 0.06 | -0.07 | -2.28 | .023 |  |
| T2 Accepting Fate | T1 Accepting Fate | 0.61 | 0.04 | 0.55 | 15.29 | < .001 | .35 |
|  | T1 Possible Identity Certainty | -0.01 | 0.07 | -0.02 | -0.72 | .471 |  |
|  | T1+ High Stakes Test Scores | -0.21 | 0.07 | -0.10 | -3.08 | .002 |  |
| T2 Optimism for the Future | T1 Optimism for the Future | 0.64 | 0.04 | 0.60 | 18.33 | < .001 | .39 |
|  | T1 Possible Identity Certainty | 0.02 | 0.02 | 0.03 | 0.89 | .375 |  |
|  | T1+ High Stakes Test Scores | 0.08 | 0.06 | 0.03 | 1.21 | .227 |  |
| T3 Possible Identity Certainty | T1 Possible Identity Certainty | 0.29 | 0.03 | 0.29 | 8.47 | < .001 | .58 |
|  | T2 Possible Identity Certainty | 0.50 | 0.03 | 0.53 | 14.93 | < .001 |  |
|  | T2+ School Placement | 0.14 | 0.07 | 0.06 | 2.08 | .038 |  |
|  | T2 Difficulty-as-Importance | 0.18 | 0.06 | 0.08 | 2.82 | .005 |  |
|  | T2 Difficulty-as-Impossibility | -0.09 | 0.05 | -0.05 | -1.68 | .093 |  |
|  | T2 Accepting Fate | 0.04 | 0.05 | 0.02 | 0.75 | .452 |  |
|  | T2 Optimism for the Future | -0.11 | 0.08 | -0.03 | -1.49 | .136 |  |
| T3 Difficulty-as-Importance | T1 Difficulty-as-Importance | 0.18 | 0.05 | 0.16 | 3.66 | < .001 | .35 |
|  | T2 Difficulty-as-Importance | 0.45 | 0.04 | 0.42 | 11.49 | < .001 |  |
|  | T2 Possible Identity Certainty | 0.07 | 0.02 | 0.15 | 4.29 | < .001 |  |
|  | T2+School Placement | 0.01 | 0.04 | 0.01 | 0.26 | .792 |  |
| T3 Difficulty-as-Impossibility | T1 Difficulty-as-Impossibility | 0.13 | 0.04 | 0.13 | 3.48 | .001 | .42 |
|  | T2 Difficulty-as-Impossibility | 0.49 | 0.04 | 0.50 | 13.65 | < .001 |  |
|  | T2 Possible Identity Certainty | -0.04 | 0.01 | -0.10 | -3.12 | .002 |  |
|  | T2+ School Placement | -0.21 | 0.04 | -0.14 | -5.36 | < .001 |  |

*(Continued)*

**Table 6.** (Continued)

| Outcome Variable | Predictor Variable | $B$ | $SE\ B$ | $\beta$ | $z$ | $p$ | $R^2$ |
|---|---|---|---|---|---|---|---|
| T3 Accepting Fate | T1 Accepting Fate | 0.14 | 0.04 | 0.13 | 3.27 | .001 | .47 |
| | T2 Accepting Fate | 0.56 | 0.04 | 0.58 | 13.92 | < .001 | |
| | T2 Possible Identity Certainty | -0.04 | 0.01 | -0.08 | -2.59 | .010 | |
| | T2+ School Placement | -0.06 | 0.04 | -0.04 | -1.42 | .156 | |
| T3 Optimism for the Future | T1 Optimism for the Future | 0.28 | 0.04 | 0.28 | 7.84 | < .001 | .41 |
| | T2 Optimism for the Future | 0.41 | 0.03 | 0.42 | 12.48 | < .001 | |
| | T2 Possible Identity Certainty | 0.01 | 0.02 | 0.03 | 0.93 | .354 | |
| | T2+ School Placement | 0.06 | 0.05 | 0.03 | 1.25 | .212 | |

$B$ = unstandardized coefficient. $SE\ B$ = standard error of unstandardized coefficient. $\beta$ = standardized coefficient.

T1 and T2 possible identity certainty scores and their T2+ school placement, students scored higher in possible identity certainty at T3 if at T2 they scored higher in difficulty-as-importance ($B = 0.18$, $z = 2.82$, $p = .005$). Their T2 difficulty-as-impossibility, accepting fate, and optimism for the future did not matter (no significant associations with T3 possible identity certainty score).

T2+ school placement and T2 possible identity certainty were the outcome variables in the sixth path. The sixth path explored the effects of T2+ school placement and T2 possible identity certainty on T3 difficulty-as-importance, difficulty-as-impossibility, optimism for the future, and accepting fate. Accounting for the effects of their T1 and T2 difficulty-as-importance scores, students scored higher in T3 difficulty-as-importance if at T2 they scored higher in possible identity certainty ($B = 0.07$, $z = 4.29$, $p < .001$); their T2+ school placement did not matter (was not significantly associated). Accounting for the effects of their T1 and T2 difficulty-as-impossibility scores, students scored lower in T3 difficulty-as-impossibility if at T2

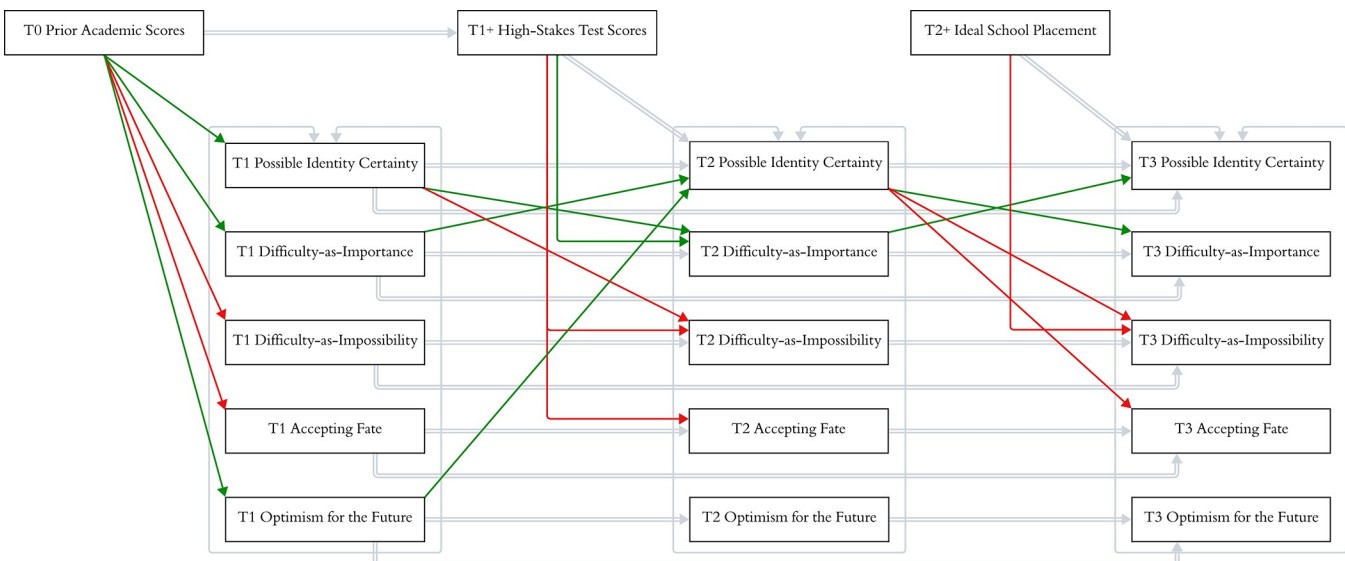

**Fig 3. Path analysis RQ2: Controlling for T0 grades and T1+ test performance, difficulty-as-importance and possible identity certainty are recursively related.** Other associations are not recursive. Table 6 details path estimates and statistics. Red lines are significantly negative, and green lines are significantly positive paths. Single-headed gray arrows are control paths. The double-headed gray arrows represent the covariances between our self and motivation constructs at each time point.

they felt more certain about attaining their academic possible identities (B = -0.04, z = -3.12, p = .002) and their T2+ school placement matched their preference ($B$ = -0.21, $z$ = -5.36, $p <$ .001). Accounting for their T1 and T2 accepting fate scores, students scored lower in T3 accepting fate if at T2, they felt more certain about attaining their academic possible identities ($B$ = -0.04, $z$ = -2.59, $p$ = .010); their T2+ school placement did not matter (was not significantly associated). Accounting for their T1 and T2 optimism for the future scores, students' T3 optimism for the future scores were not significantly associated with T2 possible identity certainty score or T2+ school placement. In sum, controlling for their prior difficulty-as-importance, difficulty-as-impossibility, and accepting fate scores, students' T2 certainty of attaining their academic possible identities mattered for their T3 difficulty-as-importance, difficulty-as-impossibility, and accepting fate scores. Their T2+ school placement was significantly associated with their T3 difficulty-as-impossibility.

## Discussion

We collected data from a large sample of 12-year-old Chinese students at three points in time during the two-month period that marked their transition from primary to secondary school. This period began a month before the school year ended, was punctuated by taking a high-stakes test, and ended with students knowing where they would attend secondary school. We predicted a recursive relationship between students' certainty of attaining their academic possible identities and their academic achievement and between their certainty of attaining their academic possible identities and their difficulty-as-importance and difficulty-as-impossibility beliefs. We explored whether this recursive relationship also holds for other self-beliefs associated with academic outcomes–optimism for the future and fatalism. We found evidence of two recursive relationships with academic possible identity certainty–one with school attainment and another with endorsing difficulty-as-importance beliefs. We also found evidence of a unidirectional relationship from academic possible identity certainty to endorsing difficulty-as-impossibility beliefs. Students who were less certain they could attain their academic possible identities were subsequently more likely to endorse difficulty-as-impossibility beliefs. The reverse was not the case, students who were surer that difficulty implies impossibility were no more or less likely to subsequently feel certain they could attain their academic possible identities. Regarding optimism for the future and accepting fate, relationships were less stable. We found associations between certainty of attaining academic possible identities and changes in optimism for the future and in accepting fate at only some time points. Our results reflect temporal changes but are based on structural equation models. We model a naturally occurring process rather than test an experimentally manipulated causal process.

### Implications for theory development

Our results are relevant to the literature on identity and school outcomes in three ways. First, our results converge with and build on cross-sectional and temporal research documenting a positive relationship between some aspects of identity and academic outcomes (for reviews, see [1, 2, 67]. Existing meta-analyses include an array of operationalizations of the self, including self-concept, self-efficacy, self-esteem, expectancy-value, among others and include an array of operationalizations of achievement [3–5]. The gap in this large body of work is that research documents an association at points in time but typically does not provide temporally lagged, multiple assessments using ecologically significant outcomes. In the current study, we address this gap. We focused on an aspect of identity that should be sensitive to fluctuation, students' certainty of attaining their academic possible identities and its lagged association with ecologically significant outcomes (grades and high-stakes test scores). We found a

recursive relationship between test scores and identity certainty even though prior grades captured much of the variability in high-stakes test scores. Our results advance the identity-achievement literature by highlighting both a specific process by which the self and academic outcomes are related (via identity certainty) and capturing a recursive process feeding into identity certainty (believing difficulty implies importance). By studying the process over multiple time points and examining how other self-beliefs feed into this recursive process, we clarify a particular part of how specific aspects of the self matter for motivation. By also exploring optimism for the future and fatalism, we rule out the alternative explanation that any positive or negative self-belief would yield the same pattern of results.

Second, our results advance research on identity-based motivation theory [6, 8, 68]. Research to date has documented that academic possible identities are associated with academic outcomes but has not separately examined the recursive processes among certainty of attaining academic possible identities, optimism for the future, accepting fate, difficulty-as-importance, and difficulty-as-impossibility. Our results imply an indirect process in which the difficulty-as-importance inferences students make matter for goal pursuit by affecting possible identities which in turn affect future-focused behavior. In line with prior research [36, 37, 43], we find that difficulty-as-importance and difficulty-as-impossibility do not function as flip sides of the same construct. While we document a recursive process between difficulty-as-importance beliefs and certainty of attaining academic possible identities, we only find evidence of a unidirectional relationship from certainty of attaining academic possible identities to difficulty-as-impossibility beliefs. Furthermore, while we found an effect of prior academic attainment on difficulty-as-impossibility beliefs, future studies are needed to understand how difficulty-as-impossibility beliefs shape aspects of school engagement beyond standardized test scores (e.g., by affecting allocation of time or attentional resources).

Third, we add to the cultural generalizability of identity-based motivation and identity-achievement research. Most research about identity-based motivation and links between identity and school outcomes focuses on the United States and to a lesser extent on other Western countries and China [10]. Our results contribute by adding to the literature including children in China [28, 69]. We document the positive effect of prior academic achievement on students' certainty of attaining their academic possible identities. Regarding difficulty-as-importance and difficulty-as-impossibility, our results are in line with prior research conducted with Western and non-Western samples [37, 43], documenting that difficulty-as-importance and difficulty-as-impossibility are related but distinct ways of making sense of difficulty. We document that difficulty-as-importance and difficulty-as-impossibility have consistent relationships with certainty of attaining academic possible identities, implying that future research would benefit from applying these constructs to understanding student motivation and outcomes. Prior studies document that difficulty-as-importance and difficulty-as-impossibility beliefs are equally accessible in China while, in contrast, difficulty-as-impossibility beliefs are more accessible in the United States [37]. Perhaps for this reason, while certainty of attaining academic possible identities consistently predicts subsequent difficulty-as-importance and difficulty-as-impossibility, only difficulty-as-importance predicts subsequent certainty of attaining academic possible identities.

## Limitations and future directions

In this section we consider limitations to generalizability given our sample, our methodological approach, and the measures we used. We also consider potential future research that could build on our findings. First, regarding our sample, a strength of our sample was that we collected data from a large sample of Chinese students at multiple time points during a critical

transition and timed data collection to yield ecologically valid results. Regarding limitations, we did not use a survey procedure to draw a representative sample of all Chinese students. The implication is that we cannot be sure that our effect sizes generalize to all Chinese students and instead consider our results an initial exploration of the recursivity prediction rather than a test of the size of effects. Our IRB-approved plan did not entail intensive outreach, so we lost some students in our summertime data collection. We used multiple imputation to impute the missing data, allowing us to move forward with a statistically powered analysis. Future research with a representative sample and IRB-approved follow-up outreach would increase the likely stability of effect sizes and confidence intervals and hence provide an estimate of the size of effects (and not just evidence that they may exist).

Second, regarding the approach, we used a quasi-experimental approach with a real school transition. A strength of our approach is that it allowed us to explore the relationships of interest with high external validity. Though ecologically valid and statistically powered, our approach was not a randomized controlled intervention. Moreover, advancements in statistical modeling (i.e., next-generation cross-lagged panel modeling) suggest caution in interpreting reciprocal effects [70, 71]. A minimum of three data points for both academic outcomes and identity is necessary for such next-generation cross-lagged analyses. We did not have a third ecologically-valid academic outcome data point. Future research could address these limitations by considering what might be a third academic outcome that could be used with a better-funded longitudinal panel, such as academic grades in the first semester of secondary school. Alternatively, future research could use a diary method to obtain additional data points during a smaller set of academic milestones than the school transition we studied.

Third, regarding our measures, a strength is that (to our knowledge) we are the first to assess certainty of attaining academic possible identities, difficulty-as-importance, difficulty-as-impossibility, optimism for the future, and accepting fate across time in a Chinese sample. As noted in the measures section, we did assess students' certainty of using effective strategies to work on their academic possible identities, a concept akin to academic self-efficacy, but this measure reduced model fit and could not be used. Future research could try to assess academic self-efficacy with differently worded measures [72]. We also did not measure how much students valued schooling, as we did not expect much variability given the high cultural normativity of valuing school [19]. Future research could explore how capturing students' expectancy-value beliefs [73] might provide insight into students' certainty of attaining their possible identities and their future-focused behavior.

## Conclusion

We find an energizing effect of feeling more certain of attaining one's academic possible identities–certainty boosts and is boosted by believing that when schoolwork feels hard to think about or do, it is probably for you (difficulty-as-importance). We infer that intervening to increase the accessibility of a difficulty-as-importance mindset may help foster a virtuous cycle in which students feel empowered to "give it their all" in high-stakes situations which can open doors to future opportunities.

## Supporting information

**S1 File. Supplemental materials.**
(DOCX)

**S2 File. PLOS Inclusivity in global research questionnaire.**
(DOCX)

## Acknowledgments

We thank the schools and children who made this paper possible and hope our results benefit educational programming and theory building. This manuscript is being submitted in partial fulfillment of requirements of the PhD of the first author, who wishes to thank each of the committee members for their helpful feedback.

## Author Contributions

**Conceptualization:** Shimin Zhu, Daphna Oyserman.

**Data curation:** Alysia Burbidge, Shimin Zhu.

**Formal analysis:** Alysia Burbidge, Sing-Hang Cheung.

**Funding acquisition:** Shimin Zhu.

**Investigation:** Shimin Zhu.

**Project administration:** Shimin Zhu, Sing-Hang Cheung.

**Resources:** Shimin Zhu.

**Validation:** Alysia Burbidge.

**Visualization:** Daphna Oyserman.

**Writing – original draft:** Alysia Burbidge, Shimin Zhu, Sing-Hang Cheung, Daphna Oyserman.

**Writing – review & editing:** Daphna Oyserman.

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
