## [Decision Letter · Decision Letter 0]

10 May 2024

PONE-D-24-06508Believing that Difficulty Signals Importance Improves School Outcomes by Bolstering School-Focused Possible IdentitiesPLOS ONE

Dear Dr. Oyserman,

Thank you for submitting your manuscript to PLOS ONE. After careful consideration, we feel that it has merit but does not fully meet PLOS ONE’s publication criteria as it currently stands. Therefore, we invite you to submit a revised version of the manuscript that addresses the points raised during the review process.

We look forward to receiving your revised manuscript.

Kind regards,

Chongzeng Bi

Academic Editor

PLOS ONE

Reviewers' comments:

Reviewer's Responses to Questions

**Comments to the Author**

1. Is the manuscript technically sound, and do the data support the conclusions?

Reviewer #1: Yes

Reviewer #2: Yes

Reviewer #3: Partly

2. Has the statistical analysis been performed appropriately and rigorously? 

Reviewer #1: Yes

Reviewer #2: Yes

Reviewer #3: Yes

3. Have the authors made all data underlying the findings in their manuscript fully available?

Reviewer #1: Yes

Reviewer #2: Yes

Reviewer #3: Yes

4. Is the manuscript presented in an intelligible fashion and written in standard English?

Reviewer #1: Yes

Reviewer #2: Yes

Reviewer #3: Yes

5. Review Comments to the Author

Reviewer #1: Review Comments

This study examined improves school outcomes by bolstering school-focused possible identities through multi-stage data collection and analysis, and the research topic chosen is of some significance, but there are some issues that need to be considered by the authors. The details are as follows:

I. Abstract

1. The keywords do not highlight the variables that are the focus of the article; please make appropriate deletions and highlight key variables.

II. Introduction

1. The author presents "A recent review of the voluminous literature on possible future selves and identities [3] documents a variety of conditions under which an accessible future self matters for behaviour". which an accessible future self matters for behaviour", from the whole introduction, there is no review of the previous literature, reflect and find contradictions, and then lead to the research purpose of this paper, that is to say, the part of the introduction is a little bit thin, logic and relevance is not strong.

2. Following up on the previous comment, can one piece of literature summarise all previous research findings? Please ask the authors to consolidate and organise the latest literature.

3. At the end of the introductory section, the authors are requested to briefly state the value and significance of the study while briefly outlining the purpose of the study in this paper.

III. Cultural context and theoretical elaboration section

1. Why do Chinese students feel academic pressure? Lack of arguments presented.

2. Based on Chinese culture, two possible student perceptions were explained: optimism and resignation. However, this kind of mentality is not only found in the Chinese student population, but is also a phenomenon that exists in all countries, and if Chinese students were selected as subjects for the study based on this statement, the rationality seems to need to be rethought. In addition, this passage still does not explain optimism and resignation enough, but only mentions them briefly through the literature, which is not deep enough, and does not really elaborate on the differences in students' academic performance due to their different perceptions of optimism and resignation to account for them.

3. The content of this part of "Identity-based motivation theory" can be properly integrated with the introduction, and part of it can be put into the introduction. At present, the content of this part of the expression gives the feeling that it is too long, please make appropriate reduction, highlighting the logic of the language, rather than a large number of words and the accumulation of literature.

IV. Methodological component

1. The author chose the primary to junior high school level, why did he choose this level of students? Reading your article, it is partly explained that only half of the students can go on to higher education, which means that there is a high degree of difficulty in this kind of higher education. However, the reality is that in the current Chinese context, only half of all promotions occur in the third year of junior high school and the first year of high school.

2. there are large regional variations in education in China, and the regional locations of the sample schools taken by the researcher should be described.

3. please put the data of demographic variables into the research methodology section through textual representation.

4. The author collected a total of 818 subjects, so how many questionnaires were distributed at each stage? What was the recovery rate? How many were lost at each stage? What data were removed? What was the basis? None of this is seen in the article.

5. Sampling: the authors mention that there were 15 schools and then 5 recruitment letters were distributed and 3 schools agreed to collect data? How did the authors manage to select 5 schools out of the 15 and distribute 5 recruitment letters? This does not seem to be explained in detail in the article.

6. The authors have translated the English scale, what is the reliability and validity of the translated scale? Does it meet the requirements? The authors did not provide details on whether the scales could be used to measure students in the Chinese cultural context.

7. The authors did not report the reliability coefficients for each scale.

8. Please make appropriate deletion and integration of the three parts of "Analysis plan, Preliminary analysis, Main analysis", which can be clearly stated.

9. Please report in detail the structural and discriminant validity ( χ2/df, CFI, TLI, RMSEA, SRMR) etc. between variables.

10. For the sample description section, please make deletions; this section is also written in a way that is not consistent with the writing style of the research methods section.

11. Please make deletions from the description of the sample, which is also not in line with the writing style of the research methodology section.

V. Discussion section

1. the results of the study are not expanded upon, just repeating the results of the previous study, it is the in-depth discussion and digging into the results of the study that is the most valuable part.

2. the discussion section is rather thin, as if it were a conclusion, not a discussion.

VI. Conclusion

1.The discussion is mixed and has a conclusion, which can be abridged, and the focus is on enriching the discussion section.

Reviewer #2: I have now had a chance to read MS PLOS-D-24-06508. Foremost, I saw a great deal to like about this article. In particular, I found the focus on reciprocal pathways to be a refreshing and exciting departure from most work that focuses on unidirectional effects. I also like that this work takes a novel yet programmatic approach to understanding the dynamic links among possible self-relevant variables in an important naturalistic context. Having said that, I did have a few minor suggestions/questions to guide the authors’ revision.

1. The authors need to define key terms (possible identities, certainty) up front. As an example of where this might matter, the authors discuss both optimism and certainty. Even though I agree that there is a meaningful difference between these two constructs, this distinction may elude naïve readers without clear definitions of both. In fact, it was only in the methods section that I was able to discern that the authors were referring to dispositional and not situated optimism (see Armor and Taylor, 1998). Thus, I would recommend clearly defining variables up front, particularly every variable represented in your path model.

2. Also, does the attainment of one’s ideal school placement represent the presumed “possible identity” here? If so, then, what exactly does the Time 3 certainty index represent given it is no longer about a “possible” but, instead, an already attained (or unattained) identity.

3. Although the results are impressive, I was wondering what objective outcomes that things like T3 possible identity certainty might predict long-term.

4. Why are Tables 6-7 included in the text and not with the rest of the tables at the end?

5. Although they focused on links with the MC inference of “difficulty as importance”, I wondered what the predictions and findings would be for the MC inference of “difficulty as improvement”?

Reviewer #3: Overall, the strength of the paper is that it tried to examine the recursive process of two types of difficult mindsets (important vs. impossibility), school-focused possible identities, and school outcomes over a short longitudinal period (two-month study) with a relatively big sample size (N= 818). The downsides are the generalizability of the findings, given that the sample is Chinese students, and the presentation of research questions, results, and interpretations throughout the paper. The paper needs a major conceptual and structural revision in order to contribute to the current literature.

1. Since the research context is situated in China, it is hard to make a global claim worldwide; citing other countries' similar findings doesn't convince the generalizability of the study findings (Check the High-stakes school transitions in China and around the world section).

2. Missing in-text citations (e.g., p. 7: The first literature focuses on self-continuity, the effects of including the future self in the current self (citation)).

3. Avoid causal language: boost, bolster, improve, etc. SEM tests the relationships among variables, and many alternative models can give similar/better model fits.

4. Instead of listing a lot of findings, summarize the most critical findings in the Abstract.

5. Research questions and hypotheses are missing in the Current Study section

6. What are the main focus(es) in the Research Questions and proposed models? You may reorganize the findings based on the main focus(es), such as recursive/unidirectional and direct/indirect relationships.

7. The model fits for RMSEA (.08) and SRMR (.08) for the second model are not very good. I'd like to know about the modification index results. Also, report 90% CI for RMSEA (a common practice).

8. The results section is difficult to understand because it lists a lot of information. More guiding summaries are necessary.

9. Some sections are redundant (e.g., Sample and Method and Sampling).

10. Some sections are not very relevant to the current study (e.g., Sample Description and Table 3).

11. The figures need to include titles and be clear. I’d recommend deleting the non-significant paths and adding significant path results (i.e., standardized coefficients).

12. Table 7: I’d recommend omitting the non-significant results. Placing the predictor in the first column and the intermediate and the outcome variables in the next columns would help understand the relationship results.

13. General writing style: Revise complex sentences that include too many meanings/explanations (e.g., Abstract: Identity-based motivation theory predicts that how well students do in school and how certain they are of attaining. . . . when school tasks and goals feel hard to think about or do.).

6. PLOS authors have the option to publish the peer review history of their article (what does this mean?). If published, this will include your full peer review and any attached files.

Reviewer #1: **Yes: **Baoyan Yang

Reviewer #2: **Yes: **Patrick J. Carroll

Reviewer #3: No

---

## [Author Response · Author response to Decision Letter 0]

6 Jul 2024

Author responses are highlighted in yellow.

Academic Editor:

Journal Requirements:

Thank you. We have updated the formatting in our manuscript to adhere to the PLOS ONE style requirements. We have also edited the supplemental materials file name to be “S1_File.docx” and note it in the “Supporting information” section of the manuscript.

We completed the questionnaire on inclusivity in global research and uploaded it as “S2_File.docx.” As requested in the questionnaire’s directions, we also now include an “Inclusivity in global research” subsection in the “Methods” section.

Thank you. We now include an ethics statement in the “Methods” section.

4. Please review your reference list to ensure that it is complete and correct. If you have cited papers that have been retracted, please include the rationale for doing so in the manuscript text, or remove these references and replace them with relevant current references. Any changes to the reference list should be mentioned in the rebuttal letter that accompanies your revised manuscript. If you need to cite a retracted article, indicate the article’s retracted status in the References list and also include a citation and full reference for the retraction notice

To the best of our knowledge, nothing we cited was later retracted. We note that to address reviewer comments, especially regarding use of our measures in Chinese samples, we expanded our references.

Reviewer #1: 

Review Comments

This study examined improves school outcomes by bolstering school-focused possible identities through multi-stage data collection and analysis, and the research topic chosen is of some significance, but there are some issues that need to be considered by the authors. The details are as follows:

I. Abstract

1. The keywords do not highlight the variables that are the focus of the article; please make appropriate deletions and highlight key variables.

Thank you. We have revised the keywords with a focus on using the names of our key variables. 

II. Introduction

1. The author presents "A recent review of the voluminous literature on possible future selves and identities [3] documents a variety of conditions under which an accessible future self matters for behaviour". which an accessible future self matters for behaviour", from the whole introduction, there is no review of the previous literature, reflect and find contradictions, and then lead to the research purpose of this paper, that is to say, the part of the introduction is a little bit thin, logic and relevance is not strong.

Thank you. We revised the “Identity-based motivation theory” section so that the review of the literature is properly sub-headed and we provide more detail on this review. The review just came out (2023) and does indeed include all references to future self or possible self or possible identity and some measure of behavior. Because it is so broad, it does highlight gaps that would otherwise not be noticed. Here is the description of the methods section from the paper:

“We chose January 1985 as our starting point, just before Markus and Nurius’s (1986) seminal paper on possible selves. We searched the PsychINFO database abstracts with the terms: “future self/ves,” “possible identity/ies,” “possible self/ves,” and “mental contrastinga” and added ancestry searches and searches based on known authors. We pulled papers from our starting point through January 27, 2022. We read abstracts and then the full studies of all papers with abstracts suggesting that the authors measured or systematically shifted focus on future me or a specific possible identity and measured a future-focused behavior or intention to act. We included studies that assessed changes in physiological measures connected to health—cortisol and blood pressure but excluded studies focused solely on mapping to regions of the brain (e.g., Tanguay, Palombo, Atance, Renoult, & Davidson, 2020). Table 2 shows our yield of 101 papers describing 170 studies and 205 results, sorted by approach (possible self, self-gap, and self-continuity). As Table 2 reveals, 4 in 10 studies focused on possible selves (37.6%) and self-continuity (40.6%).”

aWe focused on behavior rather than outcomes such as affect, optimism, or depression, for which other more targeted reviews are available (e.g., Carrillo et al., 2019; Heekerens & Eid, 2021, meta-analyses of best possible self writing studies using non-behavioral dependent variables; Schubert, Eloo, Scharfen, & Morina, 2020 meta-analysis of the effect of positive and negative future selves on positive and negative affect). We included mental contrast studies that use a behavioral dependent variable if the future and present seem self-like, though not specified as being about the self. We excluded studies that combined a mental contrast with something else if the mental contrast effect could not be separated out. For example, a large number of studies combine mental contrast with writing implementation intentions, writing a set of contingencies linking cues (if) to behaviors (then), effects may be due to having implementation intentions or to mental contrast or to some combination of both (see, Cross & Sheffield, 2019 for a meta-analysis of mental contrasting with and without adding implementation intentions for health behaviors).

2. Following up on the previous comment, can one piece of literature summarise all previous research findings? Please ask the authors to consolidate and organise the latest literature.

We appreciate the comment. The review did indeed have the intention of summarizing the full literature, as noted above in the methods section we pulled from the paper. Given that the review is both current and thorough, summarizing that provides a comprehensive survey of the literature.

3. At the end of the introductory section, the authors are requested to briefly state the value and significance of the study while briefly outlining the purpose of the study in this paper.

We appreciate the comment and revised our manuscript such that the section “Gaps to be addressed” more clearly states the research gaps our study addresses.

III. Cultural context and theoretical elaboration section

1. Why do Chinese students feel academic pressure? Lack of arguments presented.

We appreciate the comment and revised our manuscript so that our “High-stakes school transitions in China and around the world” section better represents our argument and line of reasoning.

2. Based on Chinese culture, two possible student perceptions were explained: optimism and resignation. However, this kind of mentality is not only found in the Chinese student population, but is also a phenomenon that exists in all countries, and if Chinese students were selected as subjects for the study based on this statement, the rationality seems to need to be rethought. In addition, this passage still does not explain optimism and resignation enough, but only mentions them briefly through the literature, which is not deep enough, and does not really elaborate on the differences in students' academic performance due to their different perceptions of optimism and resignation to account for them.

We appreciate the comment and address it in two ways. First, we drop the culture-based label and clarify that these beliefs are culturally-available for Chinese and other students and that the literature documents it among children in China from the age of 9 years. The literature on possible or future selves includes both Western and Chinese samples so our research contributes to this body of work. Second, we expanded the newly named “Optimism and accepting fate” section to operationalize and situate each construct. 

3. The content of this part of "Identity-based motivation theory" can be properly integrated with the introduction, and part of it can be put into the introduction. At present, the content of this part of the expression gives the feeling that it is too long, please make appropriate reduction, highlighting the logic of the language, rather than a large number of words and the accumulation of literature.

We appreciate this comment and respond by providing revised headings, taking out redundancies, and adding subheadings so that the unique content of each subheading stands out. 

IV. Methodological component

1. The author chose the primary to junior high school level, why did he choose this level of students? Reading your article, it is partly explained that only half of the students can go on to higher education, which means that there is a high degree of difficulty in this kind of higher education. However, the reality is that in the current Chinese context, only half of all promotions occur in the third year of junior high school and the first year of high school.

Thank you for this question. We have revised the “High-stakes school transitions in China and around the world” section to better represent our argument and line of reasoning. Regarding the point that “only half of all promotions occur in the third year of junior high school and the first year of high school” we are not sure we understand this comment’s meaning–are you saying that even students who make it through to a good secondary school may not go further? 

2. there are large regional variations in education in China, and the regional locations of the sample schools taken by the researcher should be described.

Thank you for noting this omission. We added a section titled “Participants” and included the province, Guangdong Province. We do not name the city as the schools wished to remain anonymous and the member of our author team who collected this data assured them that the city name would not be used.

3. please put the data of demographic variables into the research methodology section through textual representation.

Thank you for this comment. We have reframed how we present the demographics of our sample in the newly named and organized “Participants” section.

4. The author collected a total of 818 subjects, so how many questionnaires were distributed at each stage? What was the recovery rate? How many were lost at each stage? What data were removed? What was the basis? None of this is seen in the article.

Thank you for noting this inadvertent omission. Assenting students for whom we had parental consent completed the questionnaires in class. Data loss across time points was due to the fact that not all students present at T1 were in the classroom at T2 and/or T3, and some were present at T2 but not T3, and the reverse.We are not using the term “recovered” because the research team members who collected questionnaires remained in the classroom while students filled them out–questionnaires were not “recovered” in the sense that we did not hand out blank questionnaires and ask for them to be dropped off or returned at a later time. We now share the number of questionnaires obtained at each time point in the “Participants” section. The percentage of questionnaires obtained at each time point was 94%, 92%, and 60%, respectively. 

5. Sampling: the authors mention that there were 15 schools and then 5 recruitment letters were distributed and 3 schools agreed to collect data? How did the authors manage to select 5 schools out of the 15 and distribute 5 recruitment letters? This does not seem to be explained in detail in the article.

We include this detail in the first paragraph of the newly named and organized “Procedure” section: “To obtain an adequate sample size, we recruited three schools by sending recruitment letters one by one to randomly selected schools, sending five letters in total.” 

6. The authors have translated the English scale, what is the reliability and validity of the translated scale? Does it meet the requirements? The authors did not provide details on whether the scales could be used to measure students in the Chinese cultural context.

We thank you for this comment. In the Measures section (Table 2), we now share the Cronbach's alpha reliability coefficients for each of our measures (values ranged from .79 to .93) and note the initial Chinese use of scales that have already been used in China. We also added a Construct Validity section to our Results. 

7. The authors did not report the reliability coefficients for each scale.

We thank you for alerting us, we now share this in the Measures section, Table 2. 

8. Please make appropriate deletion and integration of the three parts of "Analysis plan, Preliminary analysis, Main analysis", which can be clearly stated.

Thank you for this comment. We have now deleted the subheadings under “Analysis plan” and integrated the contents into one section.

9. Please report in detail the structural and discriminant validity ( χ2/df, CFI, TLI, RMSEA, SRMR) etc. between variables.

Thank you for this comment. We detail the above requested measures of fit in Supplemental Materials Tables S4 and S5 and summarize the results of these analyses in more detail in the newly reorganized Construct Validity subsection of Preliminary Results. 

10. For the sample description section, please make deletions; this section is also written in a way that is not consistent with the writing style of the research methods section.

Thank you for this comment. We revised to move this descriptive information to the Participants subheading in Supplemental Materials. We collected this descriptive information to provide a snapshot of the children in our study particularly for readers not familiar with China. 

11. Please make deletions from the description of the sample, which is also not in line with the writing style of the research methodology section.

Thank you, see above, where we believe we address the issue raised here.

V. Discussion section

1. the results of the study are not expanded upon, just repeating the results of the previous study, it is the in-depth discussion and digging into the results of the study that is the most valuable part.

We appreciate your comment. As revised, in the first paragraph of our Discussion, we summarize our results as is common in the social and developmental psychology literature as a way of orienting the reader to what we found. Then we connect our results to the broader literature and highlight the implications of our results.

2. the discussion section is rather thin, as if it were a conclusion, not a discussion.

We appreciate your comment. As revised, we provide a more detailed connection between our results and the broader literature and highlight the implications of our results.

VI. Conclusion

1.The dis

---

## [Decision Letter · Decision Letter 1]

19 Jul 2024

Believing that difficulty signals importance improves school outcomes by bolstering academic possible identities, a recursive analysis

PONE-D-24-06508R1

Dear Professor Oyserman,

I am pleased to inform you that your manuscript has been judged scientifically suitable for publication and will be formally accepted for publication once it meets all outstanding technical requirements.

Kind regards,

Chongzeng Bi

Academic Editor

PLOS ONE

Additional Editor Comments (optional):

Reviewers' comments:

Reviewer's Responses to Questions

**Comments to the Author**

1. If the authors have adequately addressed your comments raised in a previous round of review and you feel that this manuscript is now acceptable for publication, you may indicate that here to bypass the “Comments to the Author” section, enter your conflict of interest statement in the “Confidential to Editor” section, and submit your "Accept" recommendation.

Reviewer #1: All comments have been addressed

Reviewer #2: All comments have been addressed

2. Is the manuscript technically sound, and do the data support the conclusions?

Reviewer #1: Yes

Reviewer #2: Yes

3. Has the statistical analysis been performed appropriately and rigorously? 

Reviewer #1: N/A

Reviewer #2: Yes

4. Have the authors made all data underlying the findings in their manuscript fully available?

Reviewer #1: No

Reviewer #2: Yes

5. Is the manuscript presented in an intelligible fashion and written in standard English?

Reviewer #1: Yes

Reviewer #2: Yes

6. Review Comments to the Author

Reviewer #1: (No Response)

Reviewer #2: I have now had the opportunity to read the revision of MS PONE-D-24-06508R1. The authors have adequately addressed my concerns and this paper is now ready for publication.

7. PLOS authors have the option to publish the peer review history of their article (what does this mean?). If published, this will include your full peer review and any attached files.

Reviewer #1: No

Reviewer #2: **Yes: **Patrick Carroll

---

## [Editor Report · Acceptance letter]

13 Aug 2024

PONE-D-24-06508R1 

PLOS ONE

Dear Dr. Oyserman, 

I'm pleased to inform you that your manuscript has been deemed suitable for publication in PLOS ONE. Congratulations! Your manuscript is now being handed over to our production team.

Kind regards, 

on behalf of

Professor Chongzeng Bi 

Academic Editor

PLOS ONE